# Effects of Different Defatting Methods of Black Soldier Fly (*Hermetia illucens*) Larvae Meal on the Metabolic Energy and Nutrient Digestibility in Young Laying Hens

**DOI:** 10.3390/ani14172521

**Published:** 2024-08-30

**Authors:** Yizhen Xin, Meng Xu, Lifei Chen, Guiying Wang, Wenjing Lu, Ziqi Liu, Rongsheng Shang, Yifan Li, Zhuoya Wang, Haoyang Sun, Lusheng Li

**Affiliations:** 1School of Agricultural Science and Engineering, Liaocheng University, Liaocheng 252000, China; 2210190117@stu.lcu.edu.cn (Y.X.); 2210190104@stu.lcu.edu.cn (M.X.); chenlifei@lcu.edu.cn (L.C.); wangguiying@lcu.edu.cn (G.W.); 2021405728@stu.lcu.edu.cn (W.L.); 2021405721@stu.lcu.edu.cn (Z.L.); srs19990728@163.com (R.S.); 2022404118@stu.lcu.edu.cn (Y.L.); 2022404149@stu.lcu.edu.cn (Z.W.); 2023405027@stu.lcu.edu.cn (H.S.); 2Shandong Province Engineering Research Center of Black Soldier Fly Breeding and Organic Waste Conversion, Liaocheng University, Liaocheng 252000, China

**Keywords:** black soldier fly larvae meal, metabolic energy, nutrient digestibility, laying hens

## Abstract

**Simple Summary:**

Defatting can be performed mechanically by cutting the frozen insect larvae and then pressing them to enable the leakage of intracellular fat, or chemically using petroleum ether extraction of the insect meal. This study aimed to investigate the effects of different defatting methods of black soldier fly (*Hermetia illucens*) larvae meal (BSFM) on the metabolic energy and nutrient digestibility in laying hens. The results show that both defatting methods of BSFM had no adverse effects on the metabolic energy and nutrient digestibility in young laying hens, but pressed black soldier fly meal (BSFMp) demonstrated better effects on the digestibility of metabolic energy and nutrients in the feed for young laying hens.

**Abstract:**

This study aimed to investigate the effects of different defatting methods of black soldier fly (*Hermetia illucens*) larvae meal (BSFM) on the metabolic energy and nutrient digestibility in laying hens. Sixty young laying hens (Hy-Line W-36) aged 63 days were randomly divided into two groups (G1 and G2), each with five replicates of six hens housed in individual cages. Group G1 was fed 25% pressed black soldier fly meal (BSFMp) and 75% basal diet, and Group G2 was fed 25% extracted black soldier fly meal (BSFMe) and a 75% basal diet. Both diets included 5 g/kg chromium oxide as an external marker. A 7-day preliminary trial was followed by a 4-day experimental period. The results indicate that pressing and extracting significantly affected the digestibility of crude fat and total energy in BSFM, with BSFMp showing significantly higher crude fat digestibility than BSFMe. Similarly, total energy digestibility was also significantly higher in BSFMp. However, there were no significant differences in dry matter, organic matter, and crude protein digestibility between the two processing methods. The apparent metabolic energy values of BSFMp and BSFMe were 16.34 and 12.41 MJ/kg, respectively, showing a significant difference. The nitrogen-corrected metabolic energy values were 15.89 MJ/kg in BSFMp and 11.93 MJ/kg in BSFMe, indicating a highly significant difference. The digestibility of arginine and leucine in BSFMp was significantly higher than in BSFMe, while differences in lysine, cystine, threonine, tryptophan, and isoleucine were not significant. In conclusion, both defatting methods of BSFM had no adverse effects on the metabolic energy and nutrient digestibility in young laying hens, but BSFMp demonstrated better effects on the digestibility of metabolic energy and nutrients in the feed for young laying hens.

## 1. Introduction

The increases in global population and economic development have heightened the demand for animal-derived food products, which in turn has propelled the rapid expansion of the livestock industry. Feed costs account for over 60% of the total production costs in livestock farming, with protein being the most crucial nutrient in animal feed. The price of protein feeds is the most significant factor affecting feed costs [1]. The scarcity of high-quality protein sources, particularly soybean meal (SBM) and fishmeal [2], poses a significant challenge to the development of the livestock industry. Developing new, alternative protein feed resources and reducing dependence on SBM are the primary approaches to addressing this issue. Black soldier flies (*Hermetia illucens*), known for their rapid reproduction, broad diet, high bioconversion rate, minimal space requirements for cultivation, and no competition with human food resources, are among the most promising animal protein feed resources currently available [3]. Black soldier fly larval meal (BSFM) contains 40–48% crude protein and 25–40% crude fat, along with a rich profile of amino acids and mineral elements, making it a high-quality protein feed [4,5].

As a novel protein feed, BSFM is commonly used in poultry and livestock as a partial replacement for SBM or fishmeal in equal proportions. Chobanova et al. [5] utilized total energy data to formulate diets, and studied the impact of defatted BSFM as a substitute for SBM on the growth performance of broilers, demonstrating that defatted BSFM can replace SBM. Attia et al. [6] also assessed the effects of BSFM on broilers by substituting it for SBM in equal amounts, finding that the substitution did not affect the production performance of the broilers, although the experimental diets showed differences in metabolic energy and protein content between groups. Chu et al. [7] reported that adding 3% BSFM improved the production performance of laying hens and reduced the feed-to-gain ratio, but levels above 6% hindered growth. The results indicate that a moderate amount of BSFM is beneficial for laying hens, but excessive amounts inhibit their growth performance. These findings are consistent with reports on various livestock and aquaculture animals [8,9]. The reasons for these results are generally believed to be related to the metabolic energy and apparent nutrient digestibility of BSFM.

There is limited literature on the metabolic energy and nutrient digestibility of BSFM, with most studies focusing on broilers. Both full-fat BSFM [10,11] and defatted BSFM [12] have been documented. Matin et al. [13] employed force-feeding methods using cecectomized roosters as experimental animals to determine the nitrogen-corrected true metabolic energy of partially defatted BSFM, but the results were inconsistent, with significant discrepancies between data sets, indicating a need for further research to validate these findings. Many factors influence the determination of metabolic energy in poultry, including breed, sex, age, and environmental factors, leading to significant variability in the measured values of poultry metabolic energy [14].

Currently, research on the metabolic energy of BSFM primarily uses broilers as experimental animals, and there are no studies that compare different defatting methods. This experiment used the traditional substitution method with young laying hens as experimental animals. It evaluated the apparent metabolizable energy (AME), nitrogen-corrected apparent metabolic energy (AMEn), and amino acid digestibility of BSFM processed by different defatting methods, providing insights for the application of BSFM in livestock and poultry.

## 2. Materials and Methods

The animal experiment was approved by the Institutional Animal Care and Use Committee of Liaocheng University, Liaocheng, China. All experimental procedures involving the use of animals were conducted in compliance with the relevant laws and institutional guidelines.

### 2.1. Preparation of Defatted BSFM

Black soldier fly eggs were provided by Shandong Woneng Agricultural Technology Co., Ltd. (Liaocheng, China). The eggs were incubated at 32 °C, initially reared in bran (from Liaocheng Dingsheng feed Co., Ltd., Liaocheng, China) with a 65% moisture content for the first five days, and transferred on the sixth day to a mixture of kitchen waste slurry (sourced from the cafeteria of Liaocheng University and processed by a pulping machine from Shandong Woneng Agricultural Technology Co., Ltd., Liaocheng, China) for 10 days. After the rearing period, the larvae were separated, washed, and then heat-dried at 120 °C for 6 h. The dried larvae were divided into two portions for defatting, using two different methods as per the techniques described by Kong [15]. The pressing method involved steam-cooking at 140 °C for 30 min. Solvent extraction, the second method, occurred at 60 °C for 5 h. The resulting products were then ground to produce defatted BSFM, that is, pressed (BSFMp) and extracted (BSFMe) meal.

### 2.2. Metabolic Trial Design

The substitution method is a commonly used experimental technique for assessing the effectiveness of feed utilization by animals [16]. In this study, sixty young laying hens (Hy-Line W-36) aged 63 days were randomly divided into two groups (G1 and G2), each consisting of five replicates with six hens per replicate, housed in cages (47 cm × 55 cm × 40 cm). Group G1 was fed 25% BSFMp and 75% basal diet, while group G2 received 25% BSFMe and 75% basal diet. The diet composition and nutritional parameters are shown in Table 1 and Table 2. The metabolic trials utilized an indicator method, with all diets supplemented with 5 g/kg chromium oxide as an exogenous marker. A 7-day preliminary period was followed by a 4-day experimental period, with feeding times at 8:00 and 15:00 each day. Feces were collected 1 h after feeding. Fresh feces were collected daily, with impurities such as feathers removed, and the samples were treated with 10% dilute sulfuric acid for nitrogen fixation, then stored at −20 °C for evaluation of the total tract digestibility [12].

At 74 days of age, all trial hens were euthanized by intravenous injection of a barbiturate.

### 2.3. Nutrient Composition Analysis of Samples

Defatted BSFM, basal diet, and dried feces samples were ground (solid sample grinder XA-1), sieved through a 40-mesh screen, and analyzed for dry matter content using the oven drying method (GB/T 6435-2014 [17]). Protein content was determined by the Kjeldahl method (GB/T 6432-2018 [18]), crude fat content by the Soxhlet extraction method (GB/T 6433-2006 [19]), and crude fiber content by the acid-base method (GB/T 6434-2006 [20]). Neutral detergent fiber (NDF) content and acid detergent fiber (ADF) content were set according to the national standards (GB/T 20806-2006 [21] and NY/T 1459-2007 [22], respectively). Calcium content was determined by the potassium permanganate titration method (GB/T 6436-2018 [23]), and total phosphorus content by the molybdenum blue colorimetric method (GB/T 6437-2018 [24]). Total energy was measured using an automatic adiabatic calorimeter (model: PARR 6100, Comnpany, City, State, USA). Amino acid content was determined by an automatic amino acid analyzer (S-433D, Sykam, Germany) [25].

### 2.4. Calculation of Nutrient Digestibility and Metabolic Energy

The apparent digestibility coefficients of the total tract (ATTDC) and AME of the BSFM were calculated following the method described by Schiavone et al. (2017) [12]:ATTDC X diet = [(total X ingested − total X excreted)/total X ingested]
ATTDC X insect larvae meal = [ATTDC X of the insect larvae meal diet − (ATTDC X of the basal diet × 0.75)]/0.25
where X represents dry matter (DM), organic matter (OM), crude protein (CP), ether extract (EE), or gross energy (GE).
AME diet (MJ/kg) = [(feed intake × GE of the diet) − (excreta output × GE of the excreta)]/feed intake
AME insect larvae meal (MJ/kg) = [AME of the insect larvae meal diet − (AME of the basal diet × 0.75)]/0.25

Correction for zero nitrogen (N) retention was made using a factor of 36.54 KJ per gram N retained in the body in order to estimate the N-corrected apparent metabolizable energy (AMEn). N-retention was calculated using the following formula:N retention = [(feed intake × N diet) − (excreta output × N excreta)]/feed intake (kg)

### 2.5. Statistical Analyses

The experimental data were preliminarily sorted out by Excel (https://www.microsoft.com/en-us/microsoft-365/excel, accessed on 27 August 2024), and SPSS software (version 22.0; SPSS Inc., Chicago, IL, USA) was used to perform a *t*-test.

## 3. Results

### 3.1. Comparison of Nutrient Composition between Different Defatting Methods

The nutrient content, calcium, available phosphorus, total energy, and some essential amino acids of the two defatted BSFM are presented in Table 3. There were significant differences in the crude fat content between the two extraction methods, with BSFMp containing 15.36% and BSFMe containing 6.28%. BSFMp had lower contents of crude protein, crude fiber, crude ash, calcium, phosphorus, and some essential amino acids compared to BSFMe, while the total energy content was higher in BSFMp than in BSFMe, at 22.23 MJ/kg and 20.69 MJ/kg, respectively. Among the eight essential amino acids measured, leucine had the highest content (3.45 and 4.02%), followed by lysine (3.20 and 3.71%), and the sulfur-containing amino acids methionine (0.88 and 1.03%) and cysteine (0.42 and 0.45%) had the lowest contents.

### 3.2. Nutrient Digestibility, AME, and AMEn of BSFM Processed with Different Defatting Methods

The digestibility of nutrients and total energy, as well as the values for AME and AMEn of the two defatted BSFMs, are shown in Table 4. The defatting method significantly impacted the digestibility of EE and total energy of the BSFM, with BSFMp showing significantly higher EE digestibility than BSFMe (*p* = 0.002 < 0.01), and total energy digestibility was also significantly higher in BSFMp than in BSFMe (*p* < 0.05). However, there were no significant differences in the digestibility of DM, OM, and crude protein between BSFMp and BSFMe (*p* > 0.05). AME values were higher (*p* < 0.05) for BSFMp (16.34 MJ/kg) than for BSFMe (12.41 MJ/kg), and AMEn values were also higher (*p* < 0.005) for BSFMp (15.89 MJ/kg) than for BSFMe (11.93 MJ/kg).

### 3.3. Digestibility of Essential Amino Acids in BSFM Processed with Different Defatting Methods

The true digestibility of some essential amino acids in the two diets is presented in Table 5. The digestibility of leucine and arginine in the BSFMp diet was significantly higher than in the BSFMe diet (*p* < 0.05). Although the difference in methionine digestibility between the BSFMp and BSFMe diets was not significant (*p* > 0.05), there was an improvement of 7.59% (0.85 compared to 0.79). The differences in lysine, cysteine, threonine, tryptophan, and isoleucine digestibility were not significant (*p* > 0.05).

## 4. Discussion

Improving feed utilization efficiency is crucial for ensuring cost reduction and enhancing the sustainability of the livestock industry. The accurate assessment of the nutritional value of feed materials is vital for meeting animal nutritional needs and for precise formulation. Defatting is an essential production process for BSFM when it is used as a feed ingredient [26]. One reason is that black soldier fly oil is a high-quality fat source containing a high quantity and quality of unsaturated fatty acids [27,28], especially lauric acid, which is known for its antiviral and antibacterial properties [29]. Myristic acid, another component, is utilized in cleaning products [30,31]. Therefore, defatting black soldier flies can yield significant economic benefits. Another reason is that defatted BSFM is easier to store [32]. Without defatting, lipid oxidation occurs, leading to feed preservation issues, reduced palatability, and other problems that affect usability. Pressing, solvent extraction, and subcritical extraction are common methods used to extract fats from black soldier flies [33,34]. Pressing involves breaking the cell walls under pressure to extrude fats, while solvent extraction uses organic solvents to extract fats. Compared to solvent extraction, pressing typically results in higher residual fat contents.

At present, most of the application of black soldier fly powder in poultry is carried out by replacing soybean meal and fish meal, and there is no accurate means of evaluation of nutritional value. This study utilized the same oil extraction parameters as Kong [15], where the pressing method involved steam-cooking at a temperature of 140 °C for 30 min, and the solvent extraction method was conducted at 60 °C for 5 h. The fat residue in BSFMp was 15.36%, which was higher than in BSFMe (6.28%). SBM, a product of soybean oil extraction, is a primary protein feed ingredient for livestock and poultry. According to the literature, SBM typically contains 1.2–2.0% fat [35,36], which is considerably lower than the fat content of defatted BSFM.

Protein is the fundamental building block of life, playing a crucial role in body growth and development [37], cellular repair [38], and immune functions [39]. It is essential to include sufficient protein feed in poultry diets [40]. SBM, the most widely used protein source in poultry diets, contains high levels of protein, a balanced composition of amino acids, and good palatability, serving as the benchmark for evaluating other protein sources. In recent years, most applications of BSFM in poultry have aimed at replacing SBM [1,41,42]. However, the protein content of BSFM varies significantly due to factors such as the rearing substrate and the age of larvae [43]. In this study, the protein content of BSFM, produced by feeding black soldier fly larvae with kitchen waste and processed by pressing, was 44.68%, which is similar to the protein content of SBM, and 52.00% after solvent extraction, slightly higher than the protein content of SBM (44–47%) [39]. In terms of essential amino acids, the contents of lysine, threonine, and tryptophan in BSFM were significantly higher than those in SBM, while the methionine content was roughly the same. Considering the ideal protein composition for poultry [44], methionine was notably deficient. These results indicate that BSFM is a high-quality protein feed that can replace SBM, but methionine supplementation is necessary.

Calcium and phosphorus are critical nutritional elements in feed formulations, and their deficiency can lead to abnormal skeletal development in livestock and poultry [45], reduced feed intake [46], and pica [47]. Hence, assessing the content of calcium and phosphorus in feeds is of great importance. In this experiment, the calcium and available phosphorus contents measured in BSFMp were 5.66 and 0.97%, respectively, while in BSFMe, the contents were 6.43 and 1.13%, respectively. These results show a significant discrepancy in calcium content compared to the findings by Mahmoud et al. (0.68%) [10], although the phosphorus content (0.91%) was similar. Matin et al. [13] reported higher contents of calcium (1.34 and 2.28%) and phosphorus (1.03 and 1.11%) in full-fat BSFM samples. Moula et al. [48] reported an even higher calcium content (4.43%) in BSFM reared on horse manure, which is similar to our findings and distinctly different from other results. These differences could be attributed to differences in larval age and rearing substrate, and some reports suggest that the quantity of rearing substrate remaining in the gastrointestinal tract post-harvest and the cleanliness of the larvae’s surface are significant factors causing variations in crude ash content, among others. The calcium and phosphorus contents measured in BSFM in this study are notably higher than those reported for SBM in the literature [35,49].

Chitin is a natural high-molecular polysaccharide composed of N-acetyl-D-glucosamine linked by β-1,4-glycosidic bonds, with a structure similar to cellulose [50], and is extensively present in insect exoskeletons. Chitin possesses biological functions such as antibacterial activity [51] and modulation of immune responses [52]. However, most reports suggest that certain levels of chitin can reduce the feed intake of livestock and poultry [53,54], and inhibit the intestinal absorption of nutrients [55], thereby decreasing the absorption of fats and proteins and reducing protein digestibility [56]. Consequently, chitin in BSFM is generally considered to be an anti-nutritional factor, which will affect the digestibility of other substances.

The assessment of feed nutritional value encompasses both nutrient contents and the nutritional effects resulting from the animals’ utilization of the feeds. Feeds high in nutrient content but low in animal utilization are considered to be of poor quality [57]. The results of this study indicate that the AMEn of BSFMp is significantly higher than that of BSFMe, and the metabolic energy is also higher in BSFMp compared to BSFMe. These findings are similar to those of Schiavone et al. [12], who used broilers as test animals to measure the metabolic energy and AMEn of BSFM at different defatting levels, showing that highly defatted BSFM had significantly lower metabolic energy and AMEn compared to partially defatted BSFM. The metabolic energy levels for BSFMp and BSFMe in our study were 16.34 and 12.41 MJ/kg, respectively, which are similar to the values presented by Schiavone et al. [12] (16.25 and 11.55 MJ/kg), but noticeably lower than those of De Marco et al. [11] (17.38 MJ/kg) and Mahmoud et al. [10] (19.10 MJ/kg). This difference is primarily due to variations in BSFM fat content among the studies, which is strongly correlated with metabolic energy. A similar trend was observed in studies on SBM. Ali et al. [58] compared the metabolic energy of high-oleic soybeans, conventional full-fat soybeans, solvent-extracted SBM, and pressed SBM, finding that the AMEn values of high-oleic soybeans and conventional full-fat soybeans were significantly higher than those of solvent-extracted and pressed SBM, with solvent-extracted SBM having lower AMEn than pressed SBM, and metabolic energy results being highly correlated with fat content.

This study assessed the impacts of different defatting methods involving varying temperatures and durations on the nutrient digestibility of BSFM. The fat digestibility of BSFMp was significantly higher than that of BSFMe, corresponding with the metabolic energy data. However, there were no significant differences in the digestibility of dry matter, organic matter, and crude protein, suggesting that neither pressing nor extraction impacted the digestibility of these substances. In terms of amino acid digestibility, aside from arginine and leucine, which were significantly higher in pressed BSFM, there were no significant differences in the digestibility of lysine, methionine, threonine, and tryptophan—the first, second, third, and fourth limiting amino acids in poultry diets; nor were there significant differences for cysteine and isoleucine. Milani et al. [59] also studied the effects of different temperature treatments on nutrient digestibility in castrated male pigs fed SBM, finding that high temperatures significantly enhanced the apparent digestibility of protein and organic matter in SBM, with similar findings in other feed ingredients such as cottonseed meal [60], rapeseed meal [61], and meat meal [62].

Mathai et al. [63] assessed the amino acid digestibility in growing pigs fed SBM processed with different heat treatments, showing that amino acid digestibility at 110 °C was significantly higher than that at 150 °C; similar results were also obtained in broiler experiments [64]. The findings of this study differ from those mentioned above, and analyses suggest that SBM [65], cottonseed meal [60], and rapeseed meal [66] contain anti-nutritional factors whose activity can be destroyed by heating, thereby increasing digestibility. The anti-nutritional component in BSFM is chitin, which is not heat-sensitive, hence the differing results. Additionally, the reason why different heat treatments affect the digestibility of plant proteins’ amino acids is hypothesized to be related to the Maillard reaction [64,67], which requires the formation of covalent bonds between free amino or amino acid residues and the carbonyl groups of reducing sugars [68]. Although BSFM contains high levels of amino acids, its low sugar content does not facilitate the conditions necessary for the Maillard reaction.

## 5. Conclusions

This study investigated the nutrient composition, AME and AMEn digestibility, and the digestibility of some essential amino acids in BSFM processed by pressing and extraction. There were significant differences in the crude fat content between the two defatting methods, with BSFMp containing 15.36% and BSFMe containing 6.28%. The defatting method significantly impacted the digestibility of EE and the total energy of the BSFM. The AME values of BSFMp and BSFMe were 16.34 MJ/kg and 12.41 MJ/kg, respectively, and the difference was significant. AMEn was 15.89 MJ/kg and 11.93 MJ/kg, respectively, and the difference was highly significant. Changes in energy values were closely related to fat content. The digestibility of leucine and arginine in the BSFMp diet was significantly higher in the BSFMe diet. The defatting methods did not affect the digestibility of dry matter, organic matter, crude protein, lysine, cystine, threonine, tryptophan, or isoleucine, indicating that both BSFMp and BSFMe can serve as suitable components in the diets of laying hens, without adverse effects on their metabolic energy and nutrient digestibility.

## Figures and Tables

**Table 1 animals-14-02521-t001:** Dietary ratio of metabolic test.

Items	Basal Diet	G1	G2
BSFMp (%)	0.00	25.00	0.00
BSFMe (%)	0.00	0.00	25.00
Corn (%)	73.32	54.99	54.99
Soybean meal (%)	19.6	14.70	14.70
Rapeseed Meal (%)	2.29	1.72	1.72
Limestone (%)	2.56	1.92	1.92
CaHPO_4_ (%)	1.64	1.23	1.23
L-Lys·HCl (98%) (%)	0.00	0.00	0.00
DL-Met (%)	0.07	0.05	0.05
Vitamin Premix (%)	0.02	0.02	0.02
Trace-mineral Premix (%)	0.2	0.15	0.15
Salt (%)	0.3	0.22	0.22
Total (%)	100	100	100

BSFMp means pressed black soldier fly. BSFMe means extracted black soldier fly meal.

**Table 2 animals-14-02521-t002:** Nutrient composition of metabolic test.

Ingredients	Basal Diet	G1	G2
Crude protein (% DM)	16.01	23.21	24.96
Ether extract (% DM)	2.98	6.08	3.81
Crude fiber (% DM)	2.51	4.42	4.83
Crude ash (% DM)	5.90	6.78	7.17
Calcium (% DM)	1.42	2.48	2.65
Available Phosphorus (% DM)	0.41	0.55	0.59
Gross energy (MJ/kg DM)	13.83	15.93	15.55
Lys (% DM)	0.75	1.36	1.49
Met (% DM)	0.34	0.48	0.51
Cys (% DM)	0.37	0.38	0.39
Thr (% DM)	0.65	1.02	1.11
Trp (% DM)	0.19	0.85	0.94
Arg (% DM)	0.93	1.33	1.43
Leu (% DM)	0.61	1.32	1.46
Ile (% DM)	0.56	0.98	1.07

DM means dry matter. Lys = lysine; Met = methionine; Cys = cysteine; Thr = threonine; Trp = tryptophan; Arg = arginine; Leu = leucine; Ile = isoleucine.

**Table 3 animals-14-02521-t003:** Nutritional composition of black soldier fly meal after different defatting methods.

Ingredients	BSFMp	BSFMe
DM (%)	92.51	92.38
Crude protein (% DM)	44.68	52.00
Ether extract (% DM)	15.36	6.28
Crude fiber (% DM)	10.15	11.80
Crude ash (% DM)	9.43	10.98
Calcium (% DM)	5.66	6.43
Available Phosphorus (% DM)	0.97	1.13
Gross energy (MJ/kg DM)	22.23	20.69
Lys (% DM)	3.2	3.71
Met (% DM)	0.88	1.03
Cys (% DM)	0.42	0.45
Thr (% DM)	2.12	2.47
Trp (% DM)	2.86	3.21
Arg (% DM)	2.53	2.94
Leu (% DM)	3.45	4.02
Ile (% DM)	2.22	2.58

BSFMp means pressed black soldier fly. BSFMe means extracted black soldier fly meal. DM means dry matter. Lys = lysine; Met = methionine; Cys = cysteine; Thr = threonine; Trp = tryptophan; Arg = arginine; Leu = leucine; Ile = isoleucine.

**Table 4 animals-14-02521-t004:** Apparent digestibility coefficients of the total tract (ATTDC) of nutrients, AME and AMEn, and the partial essential amino of BSFMp and BSFMe for laying hens in the breeding period.

	BSFMp	BSFMe	SEM	*p*-Value
DM	0.65	0.62	0.02	0.241
OM	0.69	0.64	0.02	0.069
Crude protein	0.54	0.56	0.01	0.08
Ether extract	0.95	0.91	0.36	0.002
Gross energy	0.63	0.55	0.03	0.02
AME (MJ/kg DM)	16.34	12.41	0.13	0.015
AMEn (MJ/kg DM)	15.89	11.93	0.84	0.002

BSFMp means pressed black soldier fly. BSFMe means extracted black soldier fly meal. DM means dry matter. OM means organic matter.

**Table 5 animals-14-02521-t005:** Apparent digestibility coefficients of the total tract (ATTDC) of the partial essential amino of BSFMp and BSFMe for laying hens in the breeding period.

	BSFMp	BSFMe	SEM	*p*-Value
Lys	0.82	0.81	0.03	0.869
Met	0.85	0.79	0.02	0.069
Cys	0.44	0.44	0.01	0.570
Thr	0.75	0.74	0.02	0.589
Trp	0.66	0.70	0.02	0.135
Arg	0.81	0.77	0.01	0.029
Leu	0.88	0.84	0.02	0.036
Ile	0.86	0.84	0.02	0.172

Lys = lysine; Met = methionine; Cys = cysteine; Thr = threonine; Trp = tryptophan; Arg = arginine; Leu = leucine; Ile = isoleucine.

## Data Availability

The data presented in this study are available on request from the corresponding author. The data are not publicly available due to company policy.

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
