# Peer review of "Effects of Different Defatting Methods of Black Soldier Fly (Hermetia illucens) Larvae Meal on the Metabolic Energy and Nutrient Digestibility in Young Laying Hens"

_animals, 2024, doi:10.3390/ani14172521_

Round 1
Reviewer 1 Report
Comments and Suggestions for Authors
Reviewer comments
General comments
This study investigates the influence of different defatting methods on the metabolic energy and nutrient digestibility in young laying hens, utilizing two methods: pressing and extraction.
The study lacks a basal diet (corn SBM-diet) for comparison, which is necessary to estimate the adverse effect of both defatting methods on the metabolic energy and nutrient digestibility in young laying hens.
- The overall goal of including BSFM in laying hens’ diet is to reduce feed cost without negatively affecting the performance and production. Therefore, why was the production performance not tested? Notably, the crude protein percentage of the test diets ranged from 23-25%, which is higher than the recommended CP requirement for Hy-Line W-36. Reporting the production performance for laying hens is important.
- The animal ethical approval for this study needs to be declared.
- The conclusion needs improvement to better highlight the study’s importance, applications, limitations, and suggestions for future research.
- The statistical analysis is missing in the materials and methods section
- Referring to Tables 1 & 2 need to be included in the materials and methods section.
- Was there a basal diet treatment in this study? As only two dietary treatments were mentioned in the metabolic trial design.
- The method of calculating AMEn needs to be mentioned in the materials and methods section
- The titles of tables are not well presented and need to be improved
- The DM percentage should be included in Table 1
- The discussion needs enhancement to underscore the importance of this study. Comparing previous studies on BSFM for poultry would enrich the discussion.
Specific comments
- Line 17. Write out “BSFMp” in full at its first appearance
- Line 47. Is it true that there is a scarcity of high-quality protein sources, particularly SBM? If so, provide a reference for this statement.
- Line 68. Revise “laying chicks” to “laying hens throughout the manuscript
- Line 86. Change “apparent metablolic energy” to “apparent metabolizable energy” throughout the manuscript.
- Line 112. Why was feeding restricted?
- Lines 115-130. Add references to the analytical methodologies.
- Line 120. Add “test diets, and digesta samples”.
- Line 133. Was nutrient digestibility tested at the ileum level since digesta samples were collected from the ileum? If yes, correct the apparent digestibility coefficients of the total tract (ATTDC).
- Line 165. Change the heading to “Nutrient digestibility, AME and AMEn of BSFM….
- Line 253. Chitin was not measured in this study. Please revise.
- Line 255-257. Rewrite the entire sentence for clarity.
- Line 267. The sentence is unclear, please rewrite.
- Line 273-274. Delete this sentence as it is not relevant to the study.
- Line 275-276. The study does not involve varing temperature and duration. Please revise.
- Line 285-286. This is not relevant to this study. Delete or revise.
Reviewer 2 Report
Comments and Suggestions for Authors
Dear Editor,
Here is my review on the manuscript with the number 3158082-Animals
Thanks for the cooperation.
Regards

Dear Editor,
Here is my review on the manuscript with the number 3158082-Animals
Thanks for the cooperation.
Regards
Reviewer 3 Report
Comments and Suggestions for Authors
The article "Effects of Different Defatting Methods of Black Soldier Fly (Hermetia illucens) Larvae Meal on Metabolic Energy and Nutrient Digestibility in Young Laying Hens" is well written and contains sufficient information regarding the different defatting methods (mechanical and chemical). However, some parts need to be improved and explained further. Therefore, you can find my comments below.
Line 94 What type of bran is being referred to? Is it wheat bran?
Line 98 Please provide additional details on how the larvae were separated.
Line 97-98 Please provide information about the starvation procedure and the method used for inactivating the larvae.
Line 109/110 Please provide more information about the basal diet. What was its composition?
Line 119 How many replicates were used in the study?
Line 123 What factor was used in the Kjeldahl analysis?
Have you used any statistical methods? Please include this information in the article to facilitate comparison of the results.
Line 147 If you report statistical differences, please provide the statistical methods used and include the letters indicating statistical differences in Table 1.
Line 152 Arginine and cysteine are not essential amino acids.
Line 156 Please rename the title of the Table „Nutritional composition table of different defatted Black soldier fly meal“ into „Nutritional composition of black soldier fly meal after different defatting methods”
Table 1 What about other amino acids? Did you analyze additional amino acids, or were only Lys, Met, Cys, Thr, Trp, Arg, Leu, and Ile selected? If so, why were these specific amino acids chosen?
Table 2 Please separate the Table2 into two distinct tables: one for the composition of the diets and another for the nutrient composition of the diets.
Line 183 Arginine and cysteine are not essential amino acids
Line 222 If you used a 6.25 factor to calculate the protein content, please mention the potential for overestimating the protein content.
Line 253 How was the chitin content measured, and where are these results reported?
Comments on the Quality of English LanguageThe English language is generally fine, but minor editing is required.
